# Combined Computer-Aided Predictors to Improve the Thermostability of Nattokinase

**DOI:** 10.3390/foods12163045

**Published:** 2023-08-14

**Authors:** Yuan Li, Liangqi Chen, Xiyu Tang, Wenhui Zhu, Aixia Ma, Changyu Shi, Jinyao Li

**Affiliations:** 1Institute of Materia Medica, Xinjiang University, Urumqi 830017, China; liyuanstc@xju.edu.cn; 2Xinjiang Key Laboratory of Biological Resources and Genetic Engineering, College of Life Science and Technology, Xinjiang University, Urumqi 830017, China; clqxju@stu.xju.edu.cn (L.C.); txyxju@stu.xju.edu.cn (X.T.); zwhxju@163.com (W.Z.); m16699218048@163.com (A.M.); shichangyu001108@163.com (C.S.)

**Keywords:** nattokinase, thermostability prediction algorithm, site-directed mutagenesis, thermostability, molecular dynamic simulation

## Abstract

Food-derived nattokinase has strong thrombolytic activity and few side effects. In the field of medicine, nattokinase has been developed as an adjuvant drug for the treatment of thrombosis, and nattokinase-rich beverages and health foods have also shown great potential in the field of food development. At present, the poor thermostability of nattokinase limits its industrial production and application. In this study, we used several thermostability-prediction algorithms to predict nattokinase from *Bacillus mojavensis* LY-06 (AprY), and screened two variants S33T and T174V with increased thermostability and fibrinolytic activity. The t_1/2_ of S33T and T174V were 8.87-fold and 2.51-fold those of the wild type AprY, respectively, and their enzyme activities were also increased (1.17-fold and 1.28-fold, respectively). Although the thermostability of N218L was increased by 2.7 times, the fibrinolytic activity of N218L was only 73.3% of that of wild type AprY. The multiple-point mutation results showed that S33T-N218L and S33T-T174V-N218L variants lost their activity, and the T174V-N218L variant did not show any significant change in catalytic performance, while S33T-T174V increased its thermostability and activity by 21.3% and 24.8%, respectively. Although the S33T-T174V variant did not show the additive effect of thermostability, it combined the excellent transient thermostability of S33T with the better thrombolytic activity of T174V. Bioinformatics analysis showed that the overall structure of S33T and T174V variants tended to be stable, while the structure of S33T-T174V variant was more flexible. Local structure analysis showed that the increased rigidity of the active center region (positions 64–75) and the key loop region (positions 129–130, 155–163, 187–192, 237–241, and 268–270) determined the increased thermostability of all variants. In addition, the enhanced flexibility of S33T-T174V variant in the Ca1 binding region (positions 1–4, 75–82) and the peripheral region of the catalytic pocket (positions 210–216) may account for the inability to superpose its thermostability. We explored the effective strategy to enhance the thermostability of nattokinase, and the resulting variants have potential industrial production and application.

## 1. Introduction

Nattokinase was first discovered by Sumi in the traditional Japanese fermented food natto in 1987, and has been widely used in the field of health food due to its powerful thrombolytic effect [1]. The traditional thrombolytic drugs streptokinase and urokinase only dissolve the thrombus itself, while nattokinase has dual functions of direct and indirect thrombolysis, which gives it a powerful thrombolytic effect [2]. In terms of direct thrombolytic mechanism, nattokinase can hydrolyze the thrombus into amino acids and small peptides to directly achieve the thrombolytic effect; in terms of indirect thrombolytic mechanisms, nattokinase promotes the conversion of prourokinase to urokinase and plasminogen conversion by stimulating the production of tissue plasminogen activator (t-PA) by vascular endothelial cells [3,4,5]. In addition to the diverse thrombolytic mechanisms, nattokinase also shows better dose safety: Wu et al. demonstrated that no death or toxicological signs were observed in mice after oral administration of nattokinase twice in a day at the maximum concentration and maximum feeding volume; in addition, nattokinase also did not show mutagenic activity and induced chromosomal aberrations in genotoxicity studies [6]. At present, the study on the molecular level of nattokinase has been relatively complete, including the gene sequence, three-dimensional structure, transcriptional regulatory region, active site, substrate binding site, catalytic mechanism, and important enzymatic properties. [7,8]. In addition, nattokinase has been genetically engineered to be expressed in different hosts, making large-scale industrial production of nattokinase possible [9,10,11,12].

The clarification of the structure and catalytic mechanism of nattokinase and the realization of its genetic engineering expression provide technical support for its molecular modification research. Using the directed evolution technique, Cai et al. reported a variant of nattokinase containing 16 mutations, which had a 2.3-fold increase in fibrinolytic efficiency compared with wild-type nattokinase [13]; Weng et al. screened a nattokinase I31L variant with 1.9 times higher fibrinolytic efficiency by homologous sequence alignment [14]; Liu et al. improved the thrombolytic activity (Q59E) and stability (S78T/Y217K/N218D) of nattokinase by combining surface charge engineering, sequence alignment, and a design strategy based on literature reports [15]. Recently, Luo reported a nattokinase M4 (S62A/N181D/N259P/P57C/A92C) variant; compared with the wild type, the M4 variant has both higher thermostability and fibrinolytic activity [16]. Although there have been few reports of the increased thermostability of nattokinase, the high temperature sensitivity of nattokinase limits its industrial production and application; therefore, the thermostability of nattokinase needs to be further improved. In addition, many studies have shown that there is an trade-off between the catalytic activity and the stability of enzymes; thus, the improvement of the thermostability of enzymes will inevitably reduce their catalytic activity [16,17]. Therefore, the design of a new nattokinase with improved thermostability and without affecting the catalytic activity has become a key issue to be solved urgently for the industrial production and oral application of nattokinase.

Protein engineering is an effective tool to improve the thermostability of enzymes [18,19]. At present, irrational design, semi-rational design, and rational design strategies are used to modify the thermostability of enzymes [20]. Irrational design strategies (such as error-prone PCR and DNA shuffling) have disadvantages, such as a large colony screening effort and their inapplicability to slow-growing host-derived enzymes [21]. The semi-rational design strategy based on site-directed saturation mutation is more likely to produce positive results, but there is still the problem of a large mutant library [18,22]. Rational design strategy is based on computer-aided technology, which has the characteristics of a small mutation library and high screening accuracy; hence, it is an effective strategy to improve the thermostability of enzymes.

Until now, several thermostability-prediction algorithms based on machine learning have been developed to predict the effects of mutations on protein thermostability. These computational approaches can be divided into two categories: classification or induction of the data using machine learning and screening methods such as support vector machine (SVM), decision tree (DT), or random forest (RF), or using statistical or empirical methods to resolve atomic interactions or structural properties [23]. For example, MUpro, I-Mutant 3.0, and iStable 2.0 use the SVM method to predict the effect of amino acid mutation on protein thermostability based on free energy change (ΔΔG) [24,25,26]; AUTO-MUTE has a variety of functional prediction types and a variety of classification and regression models, combining RF, SVM, AdaBoostM1, and C4.5 decision trees and other algorithms [27]. The FoldX program uses empirical force fields to calculate relative free energy differences caused by changes in interactions within the mutant protein [28]; CUPSAT evaluates the amino acid environment of the mutated site by calculating the amino acid atomic potential and torsion-angle distribution, and uses its solvent accessibility and secondary structure specificity to distinguish the amino acid environment [29]. Although a variety of thermostability-prediction algorithms based on different strategies have been developed, they differ markedly in the size of the screening library [30]. Recent studies have shown that the combined use of different prediction algorithms is able to improve the accuracy of protein thermostability prediction while reducing the screening effort [31,32]. By combining three thermostability-prediction algorithms (FoldX, Rosdetta ddg_monomer program, and I-Mutant 3.0), Li et al. successfully increased the optimum temperature (10 °C) and half-life (5.5-fold at 50 °C) of Candida-rugosa-derived lipase [30]; Bi et al. combined FoldX, I-Mutant 3.0, and dDFIRE to predict the thermostability of pullulanase derived from *Bacillus thermoleovorans*, and finally obtained a variant G692M with elevated Tm value (3.8 °C) and half-life (2.1-fold at 70 °C) [18]. Taken together, we suggest that combining strategies for thermostability-prediction algorithms is able to improve the thermostability of nattokinase.

In this study, putative stable variants of *Bacillus mojavensis* LY-06 nattokinase (AprY) were identified with seven prediction algorithms (I-Mutant 3.0, FoldX, CUPSAT, AUTO-MUTE RF/SVM, iStable, Mupro, and Fireprot), and the selected variants with both activity and stability were expressed, purified, and characterized. In addition, a variety of bioinformatic analysis methods were used to analyze the reasons for the improved thermostability of the selected variants.

## 2. Materials and Methods

### 2.1. Plasmids, Strains, and Cultivation Conditions

A recombinant plasmid pET-28a-AprY (resistance to kanamycin) containing a nattokinase full peptide (signal peptide + propeptide + mature peptide, recombinant AprY, rAprY) from *Bacillus mojavensis* LY-06 was used as the expression vector [33]. *Escherichia coli* BL21 (DE3; *E. coli* BL21) was used as the expression host for rAprY [33]. Luria-Bertani (LB) medium (5 g/L yeast extract, 10 g/L tryptone, 10 g/L NaCl, pH 7.0) was used for plasmid propagation and overexpression. The Isopropyl-beta-D-thiogalactopyranoside (IPTG) was used for recombinant protein expression.

### 2.2. Prediction of Point Mutation ΔΔG Values by Different Thermostability-Prediction Algorithms

Since the primary structure of AprY is consistent with the sequence of nattokinase AprN (PDB number: 4DWW, deposition time: 27 February 2012), the 3D structure of AprN was used as the object of the thermostability-prediction algorithms [33]. (1) FoldX: The RepairPDB module was used to identify and repair residues with poor torsion angles or van der Waals force conflicts, and then the high-resolution structure of the target proteins were submitted for ΔΔG values calculation [28]. (2) CAPSAT: The ΔΔG values were calculated by submitting the PDB number of the protein structures, the position of the mutated residues, and the mutated amino acids to CAPSAT online software (http://cupsat.tu-bs.de/, accessed on 6 June 2023) [29]. (3) I-Mutant 3.0: The PDB number of the protein structures, chain tags, position of mutated residues, mutated amino acids, reaction temperature, and pH value were submitted to I-Mutant 3.0 online software (http://gpcr2.biocomp.unibo.it/cgi/predictors/I-Mutant3.0/I-Mutant3.0.cgi, accessed on 6 June 2023) to calculate the ΔΔG value [24]. (4) Mupro: The primary sequence of the proteins, position of mutated residues, and mutated amino acids were submitted to Mupro online software (http://mupro.proteomics.ics.uci.edu/, accessed on 6 June 2023) to calculate the ΔΔG value [25]. (5) AUTO-MUTE: The RF algorithm was applied under the Stability Changes (ΔΔG) page of the AUTO-MUTE online software (http://binf.gmu.edu/automute/AUTO-MUTE_Stability_ddG.html, accessed on 6 June 2023), to which the protein structure PDB numbers, mutation sites, temperature, and pH were submitted to calculate the ΔΔG value [27]. (6) iStable: The PDB number of the protein structures, chain tags, position of mutated residues, mutated amino acids, reaction temperature, and pH value were submitted to iStable online software (http://predictor.nchu.edu.tw/istable/index.php, accessed on 6 June 2023) to calculate the ΔΔG value [26]. (7) Fireprot: The PDB number of the protein structures was submitted to Fireprot online software (https://loschmidt.chemi.muni.cz/fireprotweb/, accessed on 6 June 2023) to calculate the ΔΔG value [31].

### 2.3. Site-Directed Mutagenesis and Production of Nattokinase Variants

Primers used for site-directed mutagenesis were listed in Appendix A. The mutations were introduced into the AprY gene using the pET-28a-AprY as template, forward primer X–up, reverse primer X–down, and TransStart^®®^ FastPfu Fly PCR SuperMix (Beijing TransGen Biotech Co., Ltd., Beijing, China) by PCR [34]. The PCR products were treated by DMT enzyme and then were transformed into competent cells, and the transformants were sequenced by Shanghai Sangon Biotech Co., Ltd. (Shanghai, China).

### 2.4. Bacterial Expression and Purification of rAprY and Its Variants

The expression and purification method of rAprY and its variants refers to the previous research of Li et al. [33]. *Escherichia coli* BL21 (DE3) was used as host for protein induction, and their overexpression was achieved after 20 h of incubation at 18 °C by the addition of 0.1 mM IPTG. After induction, the cells were collected by centrifugation and sonicated. The supernatant obtained by the centrifugation of cell lysates was subjected to Ni-NTA column (Invitrogene, Carlsbad, CA, USA) and DEAE Sepharose Fast Flow column (Amersham Biosciences, Piscataway, NJ, USA) for protein purification. The protein concentration was determined by the BCA protein assay reagent kit (Pierce).

### 2.5. Assay of Thrombolytic Activity

The thrombolytic activity and thermostability of the crude enzyme of each rAprY variant were determined by the fibrin plate method [33]. The thrombolytic activity of rAprY and its variants was obtained by calculating the area of the transparent circle of the fibrin plate after 18 h of incubation at 37 °C, with a urokinase standard used as control. The enzymatic properties and kinetic parameters of the purified nattokinase were determined by tetrapeptide substrate (suc-AAPF-pNA) method [15]. One unit was defined as the amount of enzyme that produced 1 μmol p-NA per minute under assay conditions.

### 2.6. Biochemical Characterization of rAprY and Its Variants

The characterization of the enzymatic properties and the determination of kinetic parameters of purified rAprY and its variants were performed according to the method of Liu et al., with minor modifications [15]. The optimum pH of thrombolytic activity was determined using suc-AAPF-pNA as a substrate in 100 mM sodium acetate buffered citrate buffer (pH 2.0–6.0), 100 mM phosphate buffered saline buffer (PBS buffer, pH 7.0–8.0), and 100 mM glycine NaOH buffer sodium bicarbonate buffer (pH 9.0–11.0). The pH stability of the enzyme was obtained by measuring the residual thrombolytic activity after placing the enzyme in buffers of different pH for 30 min on ice. The optimal temperature was detected by performing the standard assay at temperatures that ranged from 0 to 80 °C in PBS buffer (pH 8.0). The thermostability of the enzyme was obtained by measuring the residual thrombolytic activity after placing the enzyme in a water bath at different temperatures for 30 min. The half-life values (t_1/2_) of the purified nattokinase and its variants were determined by monitoring the remaining activity after incubation at 55 °C for various periods in Tris-HCl buffer (100 mM, pH 8.0). The remaining fibrinolytic activity was recorded as a percentage of the initial activity, and the data are presented as the mean ± standard deviation.

### 2.7. Determination of the Kinetic Parameters

Kinetic parameters of rAprY and its variants were determined by suc-AAPF-pNA over a concentration from 0.05 to 1 mM in 0.2 mL Tris-HCl (50 mM, pH 8.0) at 25 °C. The K_m_ and k_cat_ values were calculated by the Lineweaver–Burk method using GraphPad Prism 5.0 software (San Diego, CA, USA). The values were the means of three independent experiments.

### 2.8. Bioinformatics Analysis

The online Swiss-Model workplace (https://swissmodel.expasy.org/interactive, accessed on 6 June 2023) was used to model the protein structure of rAprY and its variants [35]. A Ramachandran plot was constructed by submitting the constructed model to the PROCHECK module in SAVES 6.0 (https://saves.mbi.ucla.edu/, accessed on 6 June 2023) to verify the accuracy of the protein model (Appendix A) [36]. The molecular dynamic simulation (MD) of every modeled protein was carried out using YASARA (Yet Another Scientific Artificial Reality Application) software [37]. In detail, the setup included an optimization of the hydrogen-bonding network [38] to increase the solute stability, and a pKa prediction to fine-tune the protonation states of protein residues at the chosen pH of 8. NaCl ions were added with a physiological concentration of 0.9%, with an excess of either Na or Cl to neutralize the cell. After steepest-descent and simulated-annealing minimizations to remove clashes, the simulation was run for 20 ns using the AMBER14 force field for the solute and TIP3P for water [39]. The cutoff was 10 Å for Van der Waals forces. Long-range electrostatics were calculated using the particle-mesh Ewald [40] method with 1.0 nm. The equations of motions were integrated with a multiple timestep of 1.25 fs for bonded interactions and 2.5 fs for non-bonded interactions at a temperature of 328.15 K and a pressure of 1 atm (NPT ensemble) using algorithms described in detail previously [41]. Several programs were chosen to run this analysis, such as macro md_run for running the simulation, md_analyze to analyze the root mean square deviation (RMSD), radius of gyration (Rg), potential energy, solvent accessible surface area (SASA), protein secondary structure content and numbers of hydro bonds, and md_analyzeres to obtain the root mean square fluctuation (RMSF) value. The enzyme structure was visualized by PymoL software. A protein-interaction calculator (http://pic.mbu.iisc.ernet.in/job.html, accessed on 6 June 2023) was used to analyze the intermolecular interactions of protein models [42].

## 3. Results and Discussion

### 3.1. Rational Design of AprY Variants

The short shelf life of nattokinase food is not conducive to storage and consumption. Drying treatment can effectively extend the shelf life of nattokinase and reduce the loss of nutrients, so it is an effective preservation method. A thermal air-drying method has the advantage of a low cost compared with other drying methods, and is often used for drying enzyme preparations. However, the thermal air-drying method requires the enzyme to have good thermal stability, while nattokinase has low thermal stability [15]. Therefore, it is very important to improve the thermal stability of nattokinase on the basis of ensuring its activity to reduce the production cost of nattokinase. Compared with the traditional directed evolution and saturation mutation, the thermostability-prediction algorithm has lower screening intensity and higher prediction accuracy. Currently, the commonly used thermostability-prediction algorithm mainly predicts the thermostability change in the mutant by calculating the Gibbs free energy difference before and after the mutation of each amino acid residue in the protein structure [43]. Since different thermostability-prediction algorithms rely on different principles of prediction algorithms, systematic errors and random errors will inevitably occur in a single algorithm. Therefore, combining the results of multiple prediction algorithms can effectively improve the prediction accuracy [30]. In this study, five commonly used thermostability-prediction algorithms with different principles (I-Mutant 3.0, FoldX, CUPSAT, AUTO-MUTE RF, iStable, and Mupro) were used to predict the thermostability changes of each mutant of AprY, and the results of these algorithms were intersected to obtain a small mutation library (N218L, N218M, H226F, S204L, S204M, R247I, S89I, S105L). In addition, to further improve the breadth of prediction, we used Fireport prediction software to predict the thermostability of AprY, and combined the prediction results (N25G, S33T, S63G, V121I, A150V, T174V, N181D, G110P, A114P) with the prediction results obtained by the combination algorithm (Figure 1, Table 1).

### 3.2. Screening for AprY Mutations Related to Improved Thermostability

In this study, we first determined the thermostability and fibrinolytic activity of the crude enzyme of each AprY variant. The recombinant plasmids were induced by IPTG in *Escherichia coli* BL21 (DE3) [33]. The induced strains were isolated and sonicated, and the centrifuge harvested supernatants were screened for thermostability and activity using the fibrin plate method. The expression of crude enzymes in all variants was determined by SDS-PAGE, and their concentrations were adjusted to be consistent. The thermostability of each variant was first obtained by calculating their residual enzyme activity after 30 min of incubation at 55 °C, and the results showed that 7 of the 18 candidate variants had different degrees of thermostability improvement (percentage of residual enzyme activity: wild type rAprY 47.0%, T174V 63.0%, S89I 49.3%, N218L 63.6%, S53I 52.3%, S33T 81.3%, S105L 64.0%, S204M 51.2%), with a positive rate of 38.9%, which proved that the combination strategy could obtain good prediction accuracy (Figure 2A). Interestingly, in addition to the S89I, S53I, S105L, and S204M variants showing a decrease in activity, the fibrinolytic activity of T174V and S33T increased by 24.6% and 12.2%, respectively, while the N218L showed similar fibrinolytic activity to wild type rAprY (Figure 2B). Many studies have shown that the improvement of enzyme activity and thermostability cannot be improved at the same time [44]. The combined strategy of thermostability prediction designed by us is able to ensure the prediction accuracy while maintaining the activity of enzyme. In summary, according to the preliminary screening results, S33T, T174V, and N218L variants with improved thermal stability and no significant decrease in fibrinolytic activity were selected for subsequent analysis.

### 3.3. Purification and Characterization of Single-Point Variants

The fibrin plate method used in the primary screening experiment can achieve mass screening of positive mutants with improved thermal stability. However, due to the long reaction time, this method cannot be used to measure the activity of pure enzymes, so it cannot be used to characterize the properties of enzymes [45]. In this study, we used the tetrapeptide substrate method, which is faster and more sensitive, to characterize the enzymatic properties of the positive results of preliminary screening [14]. To determine the effects of the three screened variants on thermostability and fibrinolytic activity, rAprY as well as three variants S33T, T174V, and N218L were purified and characterized. Variants S33T, T174V and N218L were successfully purified (Appendix A), and their enzymatic properties were detected (Figure 3). We first determined the optimal reaction temperature and thermostability of each variant. The optimal temperature was detected by performing the standard assay at temperatures that ranged from 0 to 80 °C in pH 8.0, and the thermostability of the purified variants was evaluated by measuring the half-life values (t_1/2_) at 55 °C. Although the optimal temperature of rAprY and its variants was 55 °C, the thermostability of all variants was improved to varying degrees: after 30 min of incubation at 55 °C, the residual activity of rAprY decreased rapidly to 22.4%, while that of T174V and N218L was higher than 50%. Correspondingly, the residual enzyme activity of S33T was higher than 80% after 20 min of incubation, and was still higher than 50% after 80 min of incubation (Figure 3A,B). The half-life values of S33T, T174V, and N218L were 8.87, 2.51, and 2.71 times that of rAprY, respectively, indicating that they were more stable than wild-type rAprY at high temperature (Table 2).

Studies on the optimum pH and pH stability of rAprY and its variants showed that the optimal pH and pH stability of all variants shifted to the high pH region. The optimal pH of all variants increased from 8 to 9 (Figure 3C). After 4 h of incubation at pH 11, S33T, T174V, and N218L retained 81.3%, 66.0%, and 66.3% activity, whereas rAprY was only 55.0% (Figure 3D). Studies have shown that the substitution of key amino acids with different charged properties in the catalytic center of the enzyme are able to affect the optimal pH of the enzyme by affecting the pKa value of the active center, and changing the polarity of amino acid residues on the surface of the enzyme can also affect the optimal pH of the enzyme by changing the microenvironment profile [46,47]. Surface charge analysis of the three variants revealed that S33 and N218 were located near the active center, and their corresponding variants increased the positive charge distribution of the active center, thus leading to the upregulation of the optimal pH of S33T and N218L (Appendix A). The amino acid residue 174 is not located in the active center region of rAprY, but its mutation increases the positive charge content of the amino acid residue 174 region, which may change the optimal pH of the T174V variant through the change in microenvironment (Appendix A).

The specific enzyme activities and enzyme kinetic parameters of rAprY and its variants are shown in Table 2. The specific activities of S33T and T174V variants were 17.2% and 27.5% higher than that of rAprY, respectively, while N218L variant was slightly decreased. Different nattokinase-activity-detection methods may be the main reason for the difference between the enzymatic activity of the pure N218L enzyme and its corresponding primary screening results. The determination of the enzyme kinetic parameters of the S33T and T174V variants explained the different mechanisms of the increase in specific enzyme activity. The K_m_ value of S33T decreased and the K_cat_ value increased, indicating that the S33T mutant could increase the affinity of substrate and the catalytic rate of the active center. The increased K_m_ and K_cat_ values of the T174V variant indicated that the catalytic efficiency of T174V was mainly improved by increasing the catalytic rate of the active site.

These biochemical results of purified rAprY and its variants (Figure 3 and Table 2) were mostly consistent with the results of the preliminary screening (Figure 2). In brief, (1) S33T, T174V and N218L variants exhibited unchanged optimum reaction temperature and increased thermostability; (2) The optimal pH and pH stability of all variants shifted to the alkaline region; (3) The specific enzyme activities of S33T and T174V were increased. Although the specific enzyme activity of N218L decreased, N218L variants were considered for a subsequent multi-point mutation study, considering that co-mutation with other screened mutation sites may make up for the lack of activity due to the epistatic effect of the protein [48].

### 3.4. Biochemical Characterization of rAprY Variants with Multiple-Point Mutations

Many studies have confirmed that the catalytic properties can be further improved by combining mutation sites that affect the same catalytic property [32,49]. Since S33T, T174V, and N218L variants enhanced different catalytic properties of rAprY, we performed combinational mutagenesis to further improve the catalytic properties of rAprY. Three double mutants (S33T-T174V, T174V-N218L, and S33T-N218L) and one triple mutant (S33T-T174V-N218L) were constructed and characterized. Interestingly, two combined mutants, S33T-N218L and S33T-T174V-N218L, lost enzyme activity. We speculate that amino acid residues 33 and 218 are close to the key residues Asp32 and Ser221 of the active center, so that their simultaneous mutation may inactivate the enzyme by disrupting the normal conformation of the active center. In contrast, the specific activity of S33T-T174V increased by 24.8%, while that of T174V-N218L did not change significantly. The K_m_ value of S33T-T174V was nearly doubled compared with rAprY, and the K_cat_ value increased from 2.65 to 9.64, indicating that the increased activity of S33T-T174V was due to the increased catalytic rate of the catalytic center (Table 2).

The enzymatic properties of T174V-N218L showed that the optimal reaction temperature and pH of T174V-N218L had no change compared with rAprY, and the temperature stability, pH stability, and half-life of T174V-N218L were similar to those of rAprY. Although the optimal reaction temperature, pH, and pH stability of the S33T-T174V variant were similar to those of its single-point mutant, and its t_1/2_ (19.7 min) was also relatively lower, it showed unique advantages. The specific enzyme activity of the S33T-T174V mutant was higher than that of S33T, and after incubation at 55℃ for 10 min, the residual enzyme activity (77.1%) of the S33T-T174V mutant was higher than that of T174V (74.6%). This characteristic ensured that the S33T-T174V mutant was more resistant to transient high-temperature treatment in industrial production under the premise of enhanced fibrinolytic activity (Figure 4A–C, Table 2).

The combined mutation of several mutation sites that Improve the thermostability of the enzyme is an effective way to further improve the thermostability of the enzyme [19,32]. In this study, none of the designed multi-point variants achieved any further improvement in thermostability. S33T-N218L and S33T-T174V-N218L variants were inactivated. The T174V-N218L mutant lost the advantage of thermostability. Compared with the corresponding single-point mutation, the t_1/2_ of the S33T-T174V variant was slightly increased, but it showed a balance between specific activity and short-term thermostability. Taken together, S33T, T174V, and S33T-T174V variants have the dual advantages of fibrinolytic activity and enhanced thermostability, which can be used for industrial applications.

## 4. Understanding the Mechanism for Enhanced Thermostability and Catalytic Activity of Nattokinase Variants

To clarify the mechanism of changes in thermostability and catalytic activity of each variant, MD simulations of wild type rAprY and three variants (S33T, T174V, and S33T-T174V) were performed at 328.15 K for 20 ns. The root mean square deviation (RMSD) represents the sum of all atomic deviations between the conformation at a certain time and the target conformation, which is an important basis to measure whether the system is stable; the radius of gyration (Rg) can be used to describe the change in the overall structure: the larger the change in Rg, the more expansive the system is [50]. RMSD analysis results showed that compared with rAprY (average 1.17 Å), the RMSD of S33T (average 1.08 Å) and T174V (average 1.12 Å) decreased, indicating that their structures tended to be stable (Figure 5A). None of the Rg values of these mutants changed, indicating that none of the variant systems were significantly altered (Figure 5B). Interestingly, the S33T-T174V variant showed an increased RMSD (mean 1.22 Å) without a change in Rg value, which might be related to its relatively decreased thermostability and increased activity (Figure 5A,B). The decrease in total protein energy and folding free energy is associated with the increase in protein stability [51]. We then calculated the trend of total energy of rAprY and its variants over a period of 20 ns, which showed a decrease in total energy for all variants (Figure 5C). We also calculated the solvent-accessible surface area (SASA) of each variant. Compared with rAprY (mean 41,500 Å^2^), the SASA of S33T (average 41,647 Å^2^), T174V (average 41,644 Å^2^), and S33T-T174V (average 42,314 Å^2^) variants was slightly increased, indicating that the surface area of these mutant variants’ contact with the solution was changed, which may account for their elevated fibrinolytic activity. (Figure 5D).

To further determine the effect of the altered overall structure of the variants on their thermostability and catalytic activity, we determined the changes in the composition of the secondary structure and the number of bonds formed in each variant. Previous studies have suggested that the α-helix and β-sheet determine the rigidity of the enzyme, while the more flexible loop structure affects the thermostability of the enzyme [52,53]. Ban et al. demonstrated that the increased proportion of α-helix in the secondary structure is an important reason for the increased thermostability of 1,4-α-glucan branching enzyme in glycerol solutions containing potassium/sodium ions [54]. In this study, the secondary structure composition of all three variants did not change (Table 3, Appendix A). An analysis of the changes in the number of bonds formed in each static variant showed that there was no significant change in the number of bonds formed in each variant except S33T-N218L. The S33T-N218L variant showed increased hydrophobic interactions and hydrogen bonding, which appears to contradict its limited elevation in thermostability (Table 4, Appendix A).

In summary, the overall structure of S33T and T174V variants tended to be stable compared with rAprY, but the secondary structure composition and the number of bonds formed did not change significantly. The overall structure of S33T-T174V was more active whereas the intramolecular interactions were enhanced compared with rAprY. Previous studies have shown that structural flexibility is often closely related to catalytic activity, and that improving enzyme stability will reduce its catalytic activity [17]. In this study, for S33T and T174V variants, the overall structural stability could not explain the mechanism of their increased activity. The increased number of bond formations in the S33T-T174V variant also suggested that the mechanism of its increased enzyme activity may be related to the local conformational change. Therefore, we further analyzed the local structural changes of each variant to reveal the reasons for their altered catalytic properties.

The root mean square fluctuation (RMSF) represents the flexibility of protein amino acid residues; in this study, the RMSF of the three variants was determined [55]. The results showed that the decreased RMSF regions of the three mutants were the same, including 64–75, 129–130, 155–163, 187–192, 237–241 and 268–270 amino acid residues (Figure 6). Among them, amino acid residues 64–75 were close to the active center of the enzyme and consist of α-helix and Loop containing proton receptor His64; thus, it is speculated that this region has a greater impact on catalytic activity and stability. In addition, amino acid residues 129–130, 155–163, 187–192, 237–241, and 268–270 were located in the more flexible loop, and we speculated the increased stability of these highly flexible loop regions was the key factor to increase the thermostability of the variants (Figure 7). Interestingly, the flexibility of the 247–250 amino-acid-residue region was increased in all variant structures. Luo et al. designed a nattokinase M4 variant with increased activity and stability, and molecular dynamic simulations confirmed the existence of a “flexible region shifting” effect in the M4 variant, which was presumed to be the key factor for counteracting the stability–activity trade-off [16]. Therefore, we hypothesized that similar effects were observed in S33T, T174V, and S33T-T174V variants. Moreover, the S33T-T174V variant showed increased flexibility in the region of amino acid residues 1–4, 75–82, and 210–216. The region of amino acid residues 75 to 82 is the Ca1 binding region, while the region of amino acid residues 1–4 is immediately adjacent to the Ca1 binding region. The increased flexibility of these two regions may lead to the decreased stability of its binding to Ca1, whereas Ca1 loss was seen as a result of our modeling of S33T-T174V. Since Ca1 is considered to be closely related to the thermostability of nattokinase, the enhanced flexibility of amino acid residues 1–4 and 75–82 could interfere with the further enhancement of the thermostability of the S33T-T174V variant [2]. The amino acid residues 210–216 were close to the nucleophilic residue Ser221 in the active center. The increased flexibility of this region may improve the interaction between Ser221 and the substrate and improve the catalytic efficiency of the S33T-T174V variant.

The changes in the bonding network at the mutant sites were also analyzed (Figure 8). Compared with rAprY, Thr33 of S33T had a reduced hydrogen-bonding distance with Asp60 located on the adjacent loop, resulting in an increased bond energy and a more stable local structure. The loss of Ca2 in the T174V mutant results in an altered network of hydrogen bonds of Val174 to surrounding amino acid residues, which we speculate may account for the lower-magnitude increase in thermostability in the T174V variant. Although Thr33 of the S33T-T174V variant forms an additional hydrogen bond with Lys94, the local disintegration of the Val174 hydrogen-bond network simultaneously affects its thermostability. In summary, the enhanced thermostability of S33T, T174V, and S33T-T174V variants is more related to their local structural changes. The flexibility changes in the active center region may be related to their enhanced catalytic activities. The improvement of rigidity of the key loop determines the improvement of the stability of each variant. The decreased Ca2 binding ability of T174V and S33T-T174V mutants may affect improvements in their thermostability.

## 5. Conclusions

In recent years, thrombotic diseases have been a serious threat to people’s lives and health. The number of deaths due to thrombotic diseases is nearly one-quarter of the total number of deaths in the world, and the data show an upward trend [2]. At present, clinical thrombolytic drugs have side effects such as bleeding and pain during administration, and most of them are expensive. Therefore, it is an urgent task to seek new thrombolytic drugs with high efficiency, no side effects, and a low price [2]. In recent years, there have been many studies on the thrombolytic effect of nattokinase, and nattokinase health foods have also been endless. Drying nattokinase products can effectively maintain the expiry date of the product and maintain its activity, while the relatively economical thermal air-drying method will generate transient heat, which is detrimental to the activity of nattokinase. Current research strategies for improving the stability of nattokinase include the screening of nattokinase strains, improving the fermentation process, physical embedding, and nanoparticle conjugation [56,57,58]. In addition, molecular modification using modern genetic engineering methods has gradually become an effective and economical means to improve the thermostability of nattokinase. Directed evolution and semi-rational design are commonly used methods for molecular enzyme modification, but they have the problems of high screening intensity and a low positive rate. A variety of computer-aided thermostability-prediction algorithms developed to date can effectively solve the above problems, and the combination of prediction algorithms can further increase the prediction accuracy [18,30].

In this study, two AprY variants, S33T and T174V, with enhanced thermostability and fibrinolytic activity, were successfully obtained using several computer-aided prediction algorithms. Interestingly, the combined variant S33T-T174V did not exhibit additive thermostability effects. Using bioinformatics analysis, we found the key regions where the fibrinolytic activity and thermostability of the above mutants were increased, and the reasons for the decreased thermostability of the S33T-T174V variant were also found. This study provides guidance for subsequent researchers to further enhance the catalytic properties of nattokinase, and provides excellent enzyme molecules for the food and drug development of nattokinase products, which is of great significance for the prevention and treatment of vascular embolic diseases.

## Figures and Tables

**Figure 1 foods-12-03045-f001:**
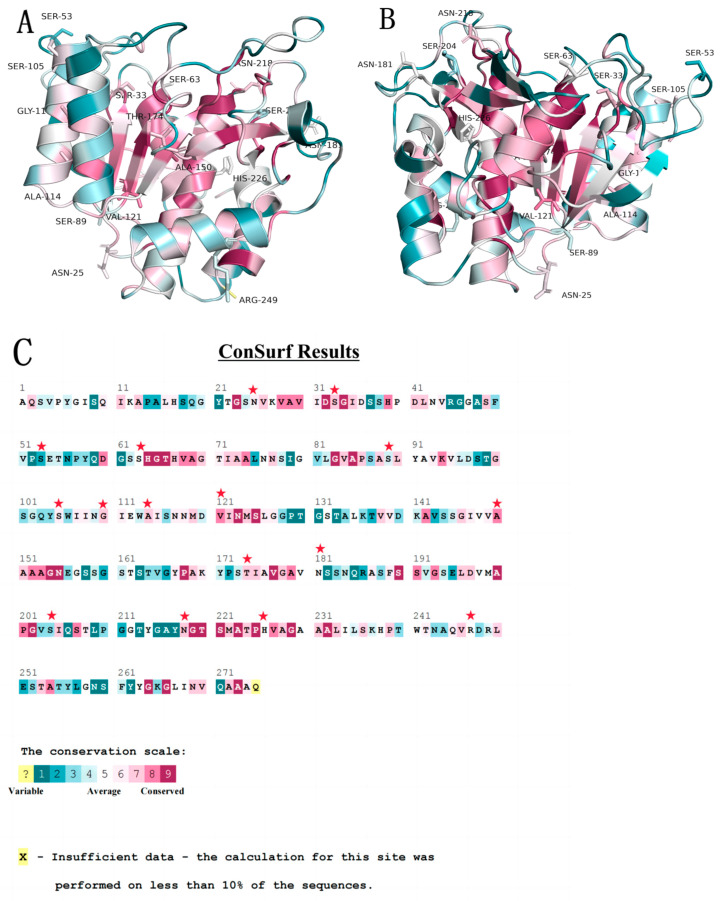
Localization of mutation candidate sites in AprY (**A**,**B**) with amino-acid-residue conservation predictions (**C**). Amino acid residues with a value of 8 or greater were considered conserved, and selected sites are marked with an asterisk in the figure.

**Figure 2 foods-12-03045-f002:**
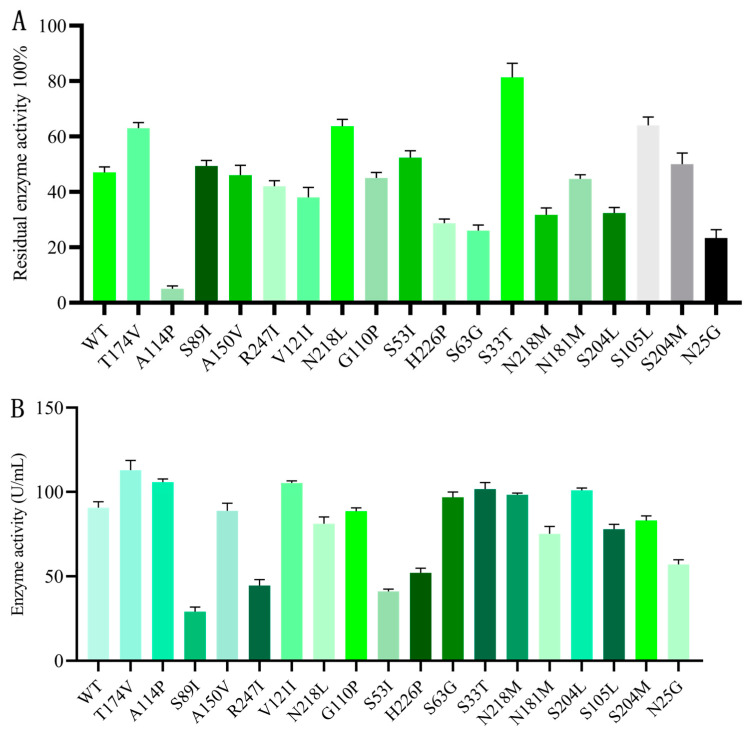
Preliminary screening of rAprY point mutations. The cells were sonicated after resuspension in Tris-HCl buffer (50 mM, pH 8.0), and the supernatant was used as a crude enzyme for screening. (**A**) Screening for thermo-resistant variants. The residual activities of rAprY and its variants were measured after the crude enzymes were incubated in pH 8.0 buffer at 55 °C for 30 min. (**B**) The fibrinolytic activity of rAprY and its variants in the supernatant.

**Figure 3 foods-12-03045-f003:**
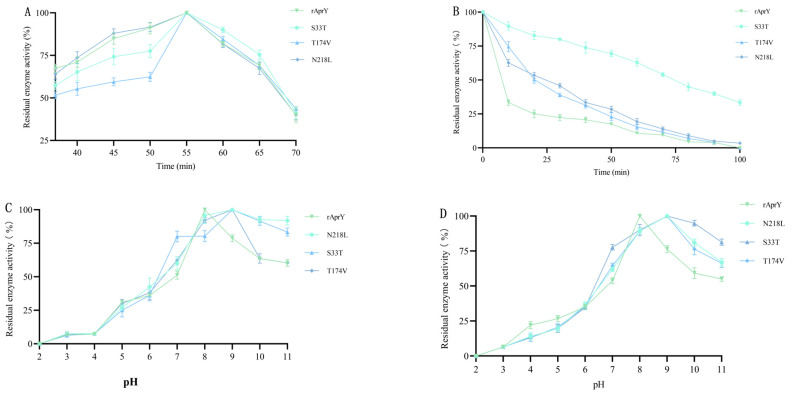
Enzymatic properties of rAprY and its variants with single-point mutations. (**A**) The optimum temperature of rAprY and single-point mutants; (**B**) The thermostability of each variant was determined by monitoring the residual activities over time (0–100 min) at 55 °C; (**C**) The optimal pH of rAprY and single-point mutants was measured using different pH conditions; (**D**) The pH stability of rAprY and its variants was determined by measuring residual activity after the enzyme was incubated in various pH conditions for 4 h at 4 °C.

**Figure 4 foods-12-03045-f004:**
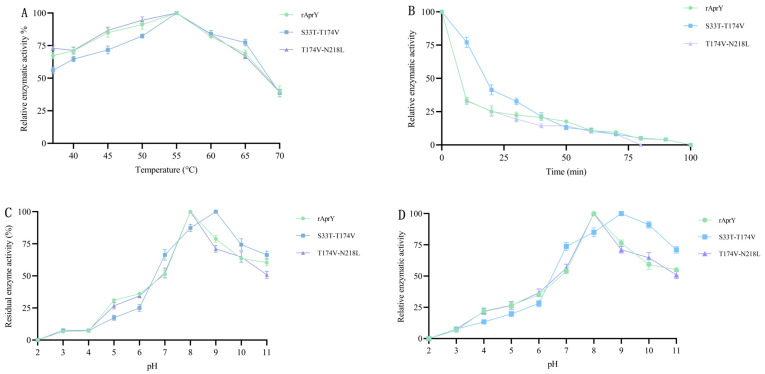
Enzymatic properties of rAprY, S33T-T174V, and T174V-N218L variants. (**A**) The optimum temperature of rAprY, S33T-T174V, and T174V-N218L variants; (**B**) The thermostability of each variant was determined by monitoring the residual activities over time (0–100 min) at 55 °C; (**C**) The optimal pH of rAprY, S33T-T174V, and T174V-N218L variants was measured using different pH conditions; (**D**) The pH stability of rAprY, S33T-T174V, and T174V-N218L variants was determined by measuring residual activity after the enzyme was incubated in various pH conditions for 4 h at 4 °C.

**Figure 5 foods-12-03045-f005:**
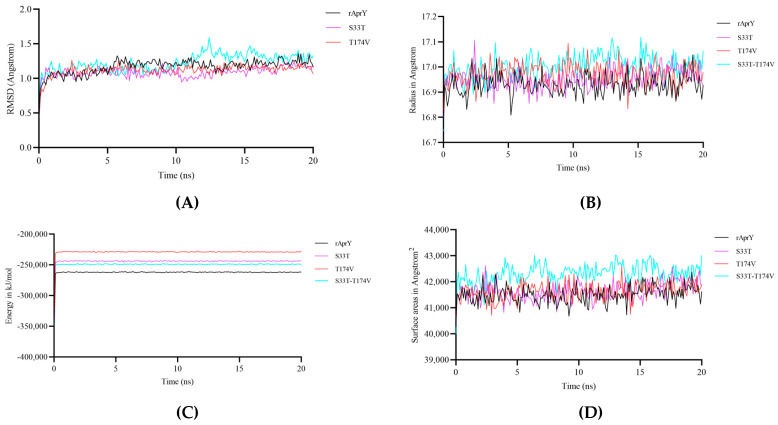
The root mean square deviation (RMSD) (**A**), Radius of gyration (Rg) (**B**), total potential energy (**C**) and solvent-accessible surface area (SASA) (**D**) of rAprY and each variant (S33T, T174V, and S33T-T1174V).

**Figure 6 foods-12-03045-f006:**
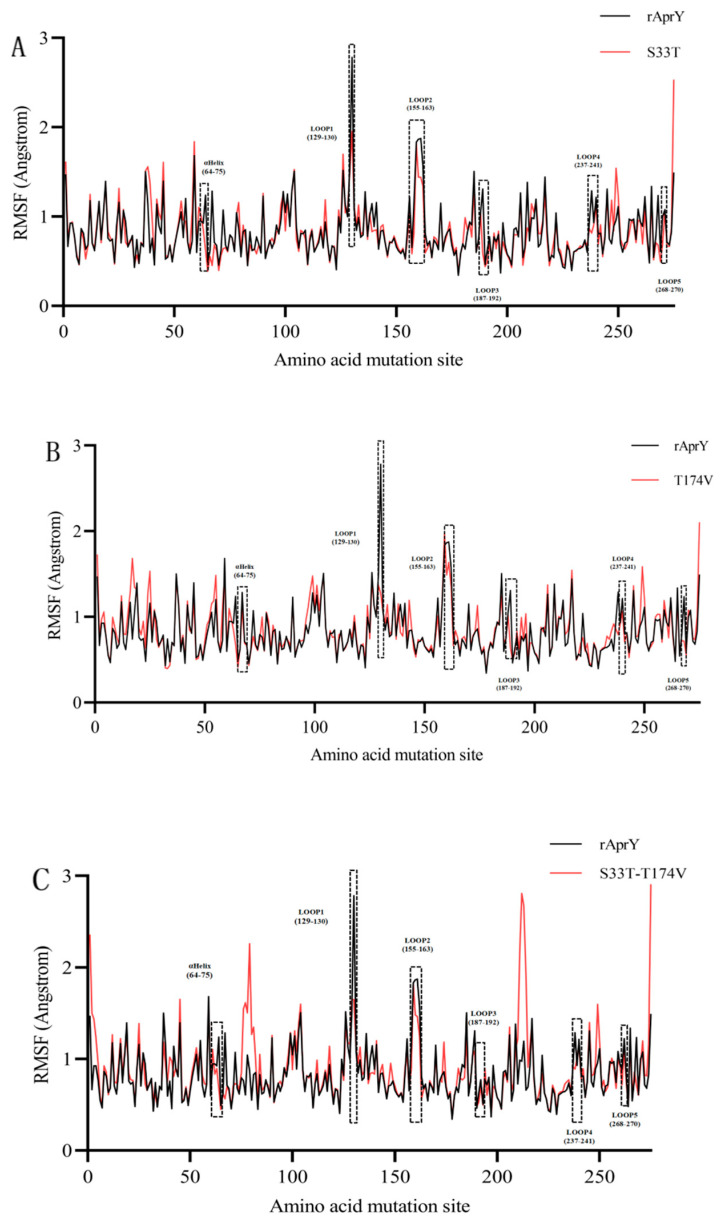
The root mean square fluctuation (RMSF) of rAprY and its variants. (**A**) rAprY and S33T; (**B**) rAprY and T174V; (**C**) rAprY and S33T-T174V.

**Figure 7 foods-12-03045-f007:**
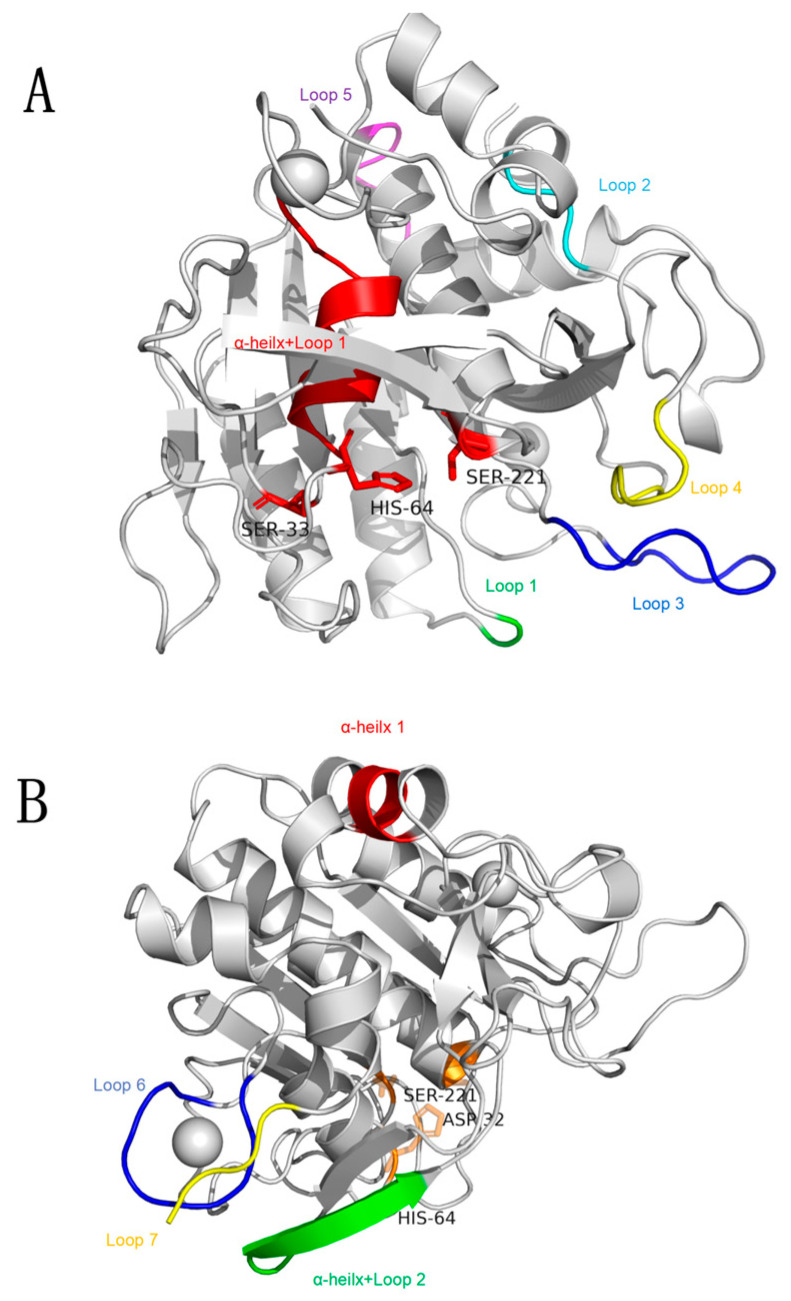
Regions of increased (**A**) and decreased (**B**) rigidity of the rAprY variant. α-helix + Loop 1: the region of amino acid residues 64–75; Loop 1: the region of amino acid residues 129–130; Loop 2: the region of amino acid residues 155–163; Loop 3: the region of amino acid residues 187–192; Loop 4: the region of amino acid residues 237–241; Loop 5: the region of amino acid residues 268–270; α-helix 1: the region of amino acid residues 247–250; α-helix + Loop 2: the region of amino acid residues 210–216; Loop 6: the region of amino acid residues 75–82; Loop 7: the region of amino acid residues 1–4.

**Figure 8 foods-12-03045-f008:**
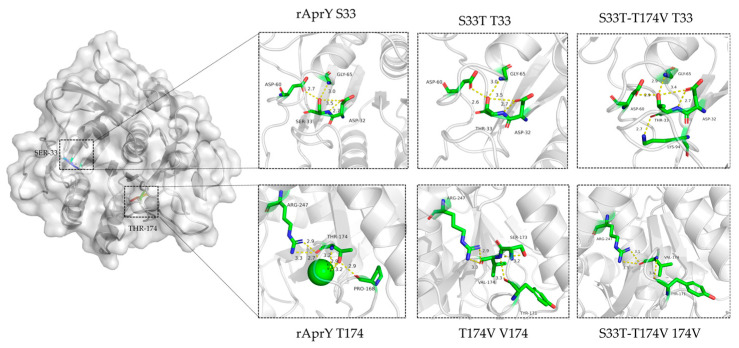
Comparison of intermolecular reactions of rAprY and its variants. (**Top**) position of Ser33; (**Bottom**) position of Thr174.

**Table 1 foods-12-03045-t001:** Stable mutants identified by different thermostability-prediction algorithms and their estimated values.

Mutation Site	FoldX (Kcal/mol)	I-Mutant 3.0 (Kcal/mol)	MUPRO (Kcal/mol)	IStable (Kcal/mol)	CUPSAT (Kcal/mol)
N218L	−0.77	0.65	1.27	0.52	0.38
N218M	−0.86	0.45	0.8	0.04	1.72
S53I	−0.77	0.2	0.5	0.21	1.34
H226F	−1.72	0.17	0.4	0.41	7.90
S204L	−0.83	0.28	0.33	0.44	0.46
S204M	−0.59	0.13	0.29	0.07	1.04
R247I	−1.03	0.25	0.1	0.16	3.52
S89I	−1.34	0.1	0.08	0.53	1.86
S105L	−0.76	0.24	0.06	0.21	0.17

**Table 2 foods-12-03045-t002:** Specific activities and kinetic parameters of wild-type rAprY and the variants.

Enzymes	rAprY	S33T	T174V	N218L	S33T-T174V
Specific activity (U/mg)	377.8 ± 25.4	442.8 ± 31.4	481.8 ± 7.1	277.0 ± 15.4	471.5 ± 23.7
Km (μmol/L)	0.48 ± 0.02	0.41 ± 0.01	0.76 ± 0.03	0.23 ± 0.01	0.97 ± 0.01
K_cat_ (S^−1^)	2.65 ± 0.07	3.89 ± 0.09	6.06 ± 0.11	1.55 ± 0.07	9.64 ± 0.11
K_d_ (×10^4^ mol^−1^·S^−1^)	5.48	9.50	8.02	6.74	9.94
t_1/2_ (min)	8.9	78.9	22.3	24.1	19.7

**Table 3 foods-12-03045-t003:** Secondary structure composition of rAprY and its variants.

Second Structure	rAprY	S33T	T174V	S33T-T174V
α-helix	30.9%	31.0%	30.61%	30.56%
β-sheet	22.2%	21.0%	21.98%	21.45%
β-turn	13.1%	13.7%	13.47%	14.54%
Coil	33.4%	33.9%	33.44%	33.02%

**Table 4 foods-12-03045-t004:** Internal interactions of rAprY and its variants.

	rAprY	S33T	T174V	S33T-T174V
Hydrophobic Interactions	258	258	265	271
Intraprotein Main Chain–Main Chain Hydrogen Bonds	279	278	279	289
Intraprotein Main Chain–Side Chain Hydrogen Bonds	127	124	121	142
Intraprotein Side Chain–Side Chain Hydrogen Bonds	63	61	61	77
Intraprotein Ionic Interactions	12	12	12	10
Intraprotein Aromatic–Aromatic Interactions	3	3	3	4
Intraprotein Aromatic–Sulphur Interactions	1	1	1	1
Intraprotein Cation–Pi Interactions	2	2	2	4

## Data Availability

The data presented in this study are available on request from the corresponding author.

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
