# Peer review of "Combined Computer-Aided Predictors to Improve the Thermostability of Nattokinase"

_foods, 2023, doi:10.3390/foods12163045_

Round 1
Reviewer 1 Report
The authors have used combination of five commonly used thermostability prediction algorithms with different principles (I-Mutant 3.0, FoldX, CUPSAT, AUTO-MUTE RF, iStable and Mupro) to improve the thermostability of the nattokinase mutant and followed by generation of a small mutation library. Therefore, tested for the invitro stability and thrombolytic activity as well as in silico MD simulations were carried out for understating the protein stability.
1. In the abstract section, the author must also add 1-2 lines on what are possible application of the thrombolytic activity of the nattokinase in food and pharmaceutical fields.
2. In the introduction section, a paragraph should be added where the applicability should be discussed with examples and the diverse mechanism of thrombolysis should be explained and how it differs from market available thrombolytic drugs.
3. In section 2.8, line number 200, during the MD simulation process, what was the solvation model used, what was the equilibration steps carried out and for what time frame should be mentioned in the materials and methods.
4. In section 4, line number 415, MD simulations were carried out for 20 ns, what was the basis for selection of the time period for MD simulation? A minimum of 100 ns simulation should be carried out in order to get biologically relevant data.
5. In figure 4, line number 404, the color code is not optimal and are confusing, can be changed to more contrasting colors. The color code for the S33T-T174V is different in figure 4A&C compared to in figure 4B&D, needs to be corrected and uniform all across.
6. The Ramachandran plot is given in supplementary figure S1 but has not been explained to what information it actually gives.
Minor editing of the language is needed to improve the flow of information in the content.
Author Response
Dear professor
We quite appreciate your favorite consideration and the reviewers’ insightful comments concerning our manuscript entitled “Combined computer-aided predictors to improve the thermostability of nattokinase” (ID: foods-2537819). Those comments are very valuable and helpful for improving the quality and readability of our paper, as well as the important guiding significance to our future researches. We have studied the comments carefully and have revised the paper exactly according to the reviewers’ comments. We hope this revision can meet with approval. The revised portions are marked in red in the paper and the main revisions corresponding to the reviewers’ comments are as follows:
Reviewer 1
- In the abstract section, the author must also add 1-2 lines on what are possible application of the thrombolytic activity of the nattokinase in food and pharmaceutical fields.
Thank you very much for your suggestion, we have added the following sentences in the abstract section: ”in the field of medicine, nattokinase has been developed as an adjuvant drug for the treatment of thrombosis, and nattokinase-rich beverages and health foods have also shown great potential in the field of food development.” We expect that this sentence will give the reader an idea of the current application of nattokinase in the pharmaceutical and food fields.
- In the introduction section, a paragraph should be added where the applicability should be discussed with examples and the diverse mechanism of thrombolysis should be explained and how it differs from market available thrombolytic drugs.
Thank you for your advice. We expanded the content of the thrombolytic mechanism, dosage and safety of nattokinase and expanded the literature. The relevant content is amended as follows:Nattokinase was first discovered by Sumi in the traditional Japanese fermented food natto in 1987, and has been widely used in the field of health food due to its powerful thrombolytic effect[1]. The traditional thrombolytic drugs streptokinase and urokinase only dissolve the thrombus itself, while Nattokinase has dual functions of direct and indirect thrombolysis, which gives it a powerful thrombolytic effect[2]. In terms of direct thrombolytic mechanism, nattokinase can hydrolyze the thrombus into amino acids and small peptides to directly achieve the thrombolytic effect; In terms of indirect thrombolytic mechanisms, nattokinase promotes the conversion of prourokinase to urokinase and plasminogen conversion by stimulating the production of tissue plasminogen activator (t-PA) by vascular endothelial cells(literature 2-4). In addition to the diverse thrombolytic mechanisms, nattokinase also showed better dose safety: Wu et al. demonstrated that no death or toxicological signs were observed in mice after oral administration of nattokinase twice in a day at the maximum concentration and maximum feeding volume; In addition, nattokinase also did not show mutagenic activity and induced chromosomal aberrations in genotoxicity studies (literature 5).
- Sumi, H.; Hamada, H.; Nakanishi, K.; Hiratani, H. Enhancement of the Fibrinolytic Activity in Plasma by Oral Administration of Nattokinases. Acta Haematol. 1990, 84, 139–143.
- Yatagai, C.; Maruyama, M.; Kawahara, T.; Sumi, H. Nattokinase-Promoted Tissue Plasminogen Activator Release from Human Cells. Pathophysiol. Haemost. Thromb. 2008, 36, 227–232.
- Urano, T.; Ihara, H.; Umemura, K.; Suzuki, Y.; Oike, M.; Akita, S.; Tsukamoto, Y.; Suzuki, I.; Takada, A. The Profibrinolytic Enzyme Subtilisin NAT Purified from Bacillus subtilis Cleaves and Inactivates Plasminogen Activator Inhibitor Type 1. J. Biol. Chem. 2001, 276, 24690–24696.
- Wu H, Wang H, Xu F, Chen J, Duan L, Zhang F. Acute toxicity and genotoxicity evaluations of Nattokinase, a promising agent for cardiovascular diseases prevention. Regul Toxicol Pharmacol. 2019 Apr;103:205-209. doi: 10.1016/j.yrtph.2019.02.006. Epub 2019 Feb 8. PMID: 30742876.
- In section 2.8, line number 200, during the MD simulation process, what was the solvation model used, what was the equilibration steps carried out and for what time frame should be mentioned in the materials and methods.
Thank you for your advice. We have refined the steps of the molecular dynamics simulations and added the corresponding references. The additions are as follows:
The setup included an optimization of the hydrogen bonding network (literature
- to increase the solute stability, and a pKa prediction to fine-tune the protonation states of protein residues at the chosen pH of 8. NaCl ions were added with a physiological concentration of 0.9%, with an excess of either Na or Cl to neutralize the cell. After steepest descent and simulated annealing minimizations to remove clashes, the simulation was run for 20 nanoseconds using the AMBER14 force field for the solute and TIP3P for water(literature7). The cutoff was 10 Å for Van der Waals forces. Long-range electrostatics were calculated using the particle mesh Ewald(literature 8)method with 1.0 nm. The equations of motions were integrated with a multiple time step of 1.25 fs for bonded interactions and 2.5 fs for non-bonded interactions at a temperature of 328.15K and a pressure of 1 atm (NPT en-semble) using algorithms described in detail previously(literature 9).
- Krieger E, Vriend G. YASARA View - molecular graphics for all devices - from smartphones to workstations. Bioinformatics. 2014 Oct 15;30(20):2981-2. doi: 10.1093/bioinformatics/btu426. Epub 2014 Jul 4. PMID: 24996895; PMCID: PMC4184264.
- Maier JA, Martinez C, Kasavajhala K, Wickstrom L, Hauser KE, Simmerling C. ff14SB: Improving the Accuracy of Protein Side Chain and Backbone Parameters from ff99SB. J Chem Theory Comput. 2015 Aug 11;11(8):3696-713. doi: 10.1021/acs.jctc.5b00255. Epub 2015 Jul 23. PMID: 26574453; PMCID: PMC4821407.
- Essmann U , Ml. B , Darden T ,et al.A SMOOTH PARTICLE MESH EWALD METHOD[J].The Journal of Chemical Physics, 1995(19):103.
- Krieger E, Vriend G. New ways to boost molecular dynamics simulations. J Comput Chem. 2015 May 15;36(13):996-1007. doi: 10.1002/jcc.23899. Epub 2015 Mar 30. PMID: 25824339; PMCID: PMC6680170.
- In section 4, line number 415, MD simulations were carried out for 20 ns, what was the basis for selection of the time period for MD simulation? A minimum of 100 ns simulation should be carried out in order to get biologically relevant data.
Thank you for your question. First, through literature reading, we believe that the determination of the time of the molecular dynamics simulation depends on whether the RMSD tends to be stable or not, the MD simulation time of 100 ns was not used in many reports in the literature, but credible results were obtained (literature 10 and 11). In general, the RMSD of protein fluctuates less than 2 Å in the MD simulation lasting 5 ns, which can be basically considered to reach equilibrium (literature 12). In this study, the RMSD value reached equilibrium (fluctuation less than 0.2 Å) after 5 ns of simulation, so we decided that the experimental data of 20 ns was sufficient for analysis.
In addition, although the data from the 20 ns MD simulation were sufficiently credible, limited by research funding and hardware facilities, it was difficult for us to complete the 100 ns MD simulation. We are solving this problem step by step and hope to get your understanding.
Relevant supporting materials are as follows:
- Liu Z, Zhao H, Han L, Cui W, Zhou L, Zhou Z. Improvement of the acid resistance, catalytic efficiency, and thermostability of nattokinase by multisite-directed mutagenesis. Biotechnol Bioeng. 2019 Aug;116(8):1833-1843. doi: 10.1002/bit.26983. Epub 2019 Apr 12. PMID: 30934114.
- Liu Z, Zhao H, Han L, Cui W, Zhou L, Zhou Z. Improvement of the acid resistance, catalytic efficiency, and thermostability of nattokinase by multisite-directed mutagenesis. Biotechnol Bioeng. 2019 Aug;116(8):1833-1843. doi: 10.1002/bit.26983. Epub 2019 Apr 12. PMID: 30934114.
- Rudrapal M, Gogoi N, Chetia D, et al. Repurposing of phytomedicine-derived bioactive compounds with promising anti-SARS-CoV-2 potential: Molecular docking, MD simulation and drug-likeness/ADMET studies. Saudi J Biol Sci. 2022 Apr;29(4):2432-2446. doi: 10.1016/j.sjbs.2021.12.018. Epub 2021 Dec 13. PMID: 34924801; PMCID: PMC8667520.
- In figure 4, line number 404, the color code is not optimal and are confusing, can be changed to more contrasting colors. The color code for the S33T-T174V is different in figure 4A&C compared to in figure 4B&D, needs to be corrected and uniform all across.
Thank you very much for your careful review of our manuscript. This is a mistake in our work. We have adjusted Figure 3 and Figure 4 again, hoping that the modified picture can meet the publication requirements in terms of color uniformity and color discrimination.
- The Ramachandran plot is given in supplementary figure S1 but has not been explained to what information it actually gives.
Thank you for your advice. We add a supplementary note to Figure S1 and point out that all protein models were of sufficient quality for subsequent studies.
Figure S1 . Ramachandran plots of rAprY and its variants: (A) wild type rAprY; (B) S33T; (C) T174V; (D)S33T-T174V. ϕ (phi) on the abscissa and ψ (psi) on the ordinate refer to the dihedral angles Ca-N-C and Cα-C-N of the protein model, respectively. Residues in the most favored regions are shown in red. Residues in additionally allowed regions are shown in yellow. All four protein models have more than 90% of the sites located in the most favored regions, confirming that these protein model residues have good confor-mational space and can be used for subsequent studies.
The relevant literature is supported as follows:
- Morris AL, MacArthur MW, Hutchinson EG, Thornton JM (1992). Stereochemical quality of protein structure coordinates. Proteins, 12, 345-364
Reviewer 2 Report
I have reviewed the paper titled “Combined computer-aided predictors to Improve the thermostability of
Nattokinase”. The paper needs modification before acceptance.
-The abstract should be more decorated, and include key findings in the abstract.
-the authors should explain the novelty of this work.
-The introduction section needs to be updated by adding related literature from the SCOPUS.
- The results and discussion should be improved.
-the resolution of Figure 1 needs to be enhanced.
There are several grammatical issues that need to be fixed.
I have reviewed the paper titled “Combined computer-aided predictors to Improve the thermostability of
Nattokinase”. The paper needs modification before acceptance.
-The abstract should be more decorated, and include key findings in the abstract.
-the authors should explain the novelty of this work.
-The introduction section needs to be updated by adding related literature from the SCOPUS.
- The results and discussion should be improved.
-the resolution of Figure 1 needs to be enhanced.
There are several grammatical issues that need to be fixed.
Author Response
Dear professor
We quite appreciate your favorite consideration and the reviewers’ insightful comments concerning our manuscript entitled “Combined computer-aided predictors to improve the thermostability of nattokinase” (ID: foods-2537819). Those comments are very valuable and helpful for improving the quality and readability of our paper, as well as the important guiding significance to our future researches. We have studied the comments carefully and have revised the paper exactly according to the reviewers’ comments. We hope this revision can meet with approval. The revised portions are marked in red in the paper and the main revisions corresponding to the reviewers’ comments are as follows:
- The abstract should be more decorated, and include key findings in the abstract.
the authors should explain the novelty of this work.
Thank you very much for your advice. We have added the application of nattokinase to the abstract, supplemented the enzymatic properties of N218L, refined the results of the combined mutations, and refined the results of the molecular dynamics simulations of S33T-T174V. The revised abstract is as follows, with the changes highlighted in red:
Food-derived nattokinase has strong thrombolytic activity and few side effects. In the field of medicine, nattokinase has been developed as an adjuvant drug for the treatment of thrombosis, and nattokinase-rich beverages and health foods have also shown great potential in the field of food development. At present, the poor thermostability of nattokinase limits its industrial production and application. In this study, we used several thermostability prediction algorithms to predict nattokinase from Bacillus mojavensis LY-06 (AprY), and screened two variants S33T and T174V with increased thermostability and fibrinolytic activity. The t1/2 of S33T and T174V were 8.87-fold and 2.51-fold of the wild type AprY, respectively, and their enzyme activities were also increased (1.17-fold and 1.28-fold, respectively). Although the thermostability of N218L was increased by 2.7 times, the fibrinolytic activity of N218L was only 73.3% of that of wild type AprY. The multiple-point mutation results showed that S33T-N218L and S33T-T174V-N218L variants lost their activity, and the T174V-N218L variant did not show any significant change in catalytic performance, while S33T-T174V increased its thermostability and activity by 21.3% and 24.8%, respectively. Although S33T-T174V variant did not show the additive effect of thermostability, it combined the excellent transient thermostability of S33T with the better thrombolytic activity of T174V. Bioinformatics analysis showed that the overall structure of S33T and T174V variants tended to be stable, while the structure of S33T-T174V variant was more flexible. Local structure analysis showed that the increased rigidity of the active center region (positions 64-75) and the key loop region (positions 129-130, 155-163, 187-192, 237-241, and 268-270) determined the increased thermostability of all variants. In addition, the enhanced flexibility of S33T-T174V variant in the Ca1 binding region (positions 1-4, 75-82) and the peripheral region of the catalytic pocket (positions 210-216) may account for the inability to superpose its thermostability. However, the increased flexibility in the active central region (positions 210-216) and the interaction region with Ca1 binding site (positions 1-4, 210-216) of the S33T-T174V variant may contribute to the decreased thermostability of the S33T-T174V variant compared with the corresponding single-point mutation. We explored the effective strategy to enhance the thermostability of nattokinase, and the resulting variants have potential industrial production and application.
- The introduction section needs to be updated by adding related literature from the SCOPUS.
Thank you for your advice. We expanded the content of the thrombolytic mechanism, dosage and safety of nattokinase and expanded the literature. The relevant content is amended as follows:
Nattokinase was first discovered by Sumi in the traditional Japanese fermented food natto in 1987, and has been widely used in the field of health food due to its powerful thrombolytic effect[1]. The traditional thrombolytic drugs streptokinase and urokinase only dissolve the thrombus itself, while Nattokinase has dual functions of direct and indirect thrombolysis, which gives it a powerful thrombolytic effect[2]. In terms of direct thrombolytic mechanism, nattokinase can hydrolyze the thrombus into amino acids and small peptides to directly achieve the thrombolytic effect; In terms of indirect thrombolytic mechanisms, nattokinase promotes the conversion of prourokinase to urokinase and plasminogen conversion by stimulating the production of tissue plasminogen activator (t-PA) by vascular endothelial cells(literature 2-4). In addition to the diverse thrombolytic mechanisms, nattokinase also showed better dose safety: Wu et al. demonstrated that no death or toxicological signs were observed in mice after oral administration of nattokinase twice in a day at the maximum concentration and maximum feeding volume; In addition, nattokinase also did not show mutagenic activity and induced chromosomal aberrations in genotoxicity studies (literature 5).
- Sumi, H.; Hamada, H.; Nakanishi, K.; Hiratani, H. Enhancement of the Fibrinolytic Activity in Plasma by Oral Administration of Nattokinases. Acta Haematol. 1990, 84, 139–143.
- Yatagai, C.; Maruyama, M.; Kawahara, T.; Sumi, H. Nattokinase-Promoted Tissue Plasminogen Activator Release from Human Cells. Pathophysiol. Haemost. Thromb. 2008, 36, 227–232.
- Urano, T.; Ihara, H.; Umemura, K.; Suzuki, Y.; Oike, M.; Akita, S.; Tsukamoto, Y.; Suzuki, I.; Takada, A. The Profibrinolytic Enzyme Subtilisin NAT Purified from Bacillus subtilis Cleaves and Inactivates Plasminogen Activator Inhibitor Type 1. J. Biol. Chem. 2001, 276, 24690–24696.
- Wu H, Wang H, Xu F, Chen J, Duan L, Zhang F. Acute toxicity and genotoxicity evaluations of Nattokinase, a promising agent for cardiovascular diseases prevention. Regul Toxicol Pharmacol. 2019 Apr;103:205-209. doi: 10.1016/j.yrtph.2019.02.006. Epub 2019 Feb 8. PMID: 30742876.
We have refined the steps of the molecular dynamics simulations and added the corresponding references. The additions are as follows:
The setup included an optimization of the hydrogen bonding network (literature
- to increase the solute stability, and a pKa prediction to fine-tune the protonation states of protein residues at the chosen pH of 8. NaCl ions were added with a physiological concentration of 0.9%, with an excess of either Na or Cl to neutralize the cell. After steepest descent and simulated annealing minimizations to remove clashes, the simulation was run for 20 nanoseconds using the AMBER14 force field for the solute and TIP3P for water(literature7). The cutoff was 10 Å for Van der Waals forces. Long-range electrostatics were calculated using the particle mesh Ewald(literature 8)method with 1.0 nm. The equations of motions were integrated with a multiple time step of 1.25 fs for bonded interactions and 2.5 fs for non-bonded interactions at a temperature of 328.15K and a pressure of 1 atm (NPT en-semble) using algorithms described in detail previously(literature 9).
- Krieger E, Vriend G. YASARA View - molecular graphics for all devices - from smartphones to workstations. Bioinformatics. 2014 Oct 15;30(20):2981-2. doi: 10.1093/bioinformatics/btu426. Epub 2014 Jul 4. PMID: 24996895; PMCID: PMC4184264.
- Maier JA, Martinez C, Kasavajhala K, Wickstrom L, Hauser KE, Simmerling C. ff14SB: Improving the Accuracy of Protein Side Chain and Backbone Parameters from ff99SB. J Chem Theory Comput. 2015 Aug 11;11(8):3696-713. doi: 10.1021/acs.jctc.5b00255. Epub 2015 Jul 23. PMID: 26574453; PMCID: PMC4821407.
- Essmann U , Ml. B , Darden T ,et al.A SMOOTH PARTICLE MESH EWALD METHOD[J].The Journal of Chemical Physics, 1995(19):103.
- Krieger E, Vriend G. New ways to boost molecular dynamics simulations. J Comput Chem. 2015 May 15;36(13):996-1007. doi: 10.1002/jcc.23899. Epub 2015 Mar 30. PMID: 25824339; PMCID: PMC6680170.
- The results and discussion should be improved.
Thank you very much for your advice. In the results and discussion section, we added the production and application of nattokinase products, the method of nattokinase activity determination, and the theoretical content of nattokinase molecular modification, respectively. As a further supplement to your suggestions, we have added the conclusion section, which gives an overview of the production and application of nattokinase, summarizes the conclusions of this study, and finally expounds the significance of the study and prospects for future research. All modified sections are marked in red and attached with added references.
- Astrup, T.; Müllertz, S. The fibrin plate method for estimating fibrinolytic activity. Arch. Biochem. Biophys. 1952, 40, 346–351.
- Weng, M.; Deng, X.; Bao, W.; Zhu, L.; Wu, J.; Cai, Y.; Jia, Y.; Zheng, Z.; Zou, G. Improving the activity of the subtilisin nattokinase by site-directed mutagenesis and molecular dynamics simulation. Biochem. Biophys. Res. Commun. 2015, 465, 580–586.
- Cheng Chen, Bi-Han Guan, Qiang Geng, Yu-Cong Zheng, Qi Chen, Jiang Pan, and Jian-He XuACS Catalysis 2023 13 (11), 7407-7416. DOI: 10.1021/acscatal.3c00503
- Dawei Ni, Shuqi Zhang, Onur Kırtel, Wei Xu, Qiuming Chen, Ebru Toksoy Öner, and Wanmeng MuJournal of Agricultural and Food Chemistry 2021 69 (44), 13125-13134
DOI: 10.1021/acs.jafc.1c04852
- Yuan, L., et al., Biotechnology, Bioengineering and Applications of Bacillus Nattokinase. Biomolecules, 2022. 12(7).

Reviewer 3 Report
The paper is well written but in my opinion it is too verbose. I believe that authors should summarize the results. A short final comment must be insert at the end of each paragraph of the experimental part. please write the conclusions in more incisive way.
Author Response
Dear professor
We quite appreciate your favorite consideration and the reviewers’ insightful comments concerning our manuscript entitled “Combined computer-aided predictors to improve the thermostability of nattokinase” (ID: foods-2537819). Those comments are very valuable and helpful for improving the quality and readability of our paper, as well as the important guiding significance to our future researches. We have studied the comments carefully and have revised the paper exactly according to the reviewers’ comments. We hope this revision can meet with approval. The revised portions are marked in red in the paper and the main revisions corresponding to the reviewers’ comments are as follows:
The paper is well written but in my opinion it is too verbose. I believe that authors should summarize the results. A short final comment must be insert at the end of each paragraph of the experimental part. please write the conclusions in more incisive way.
3.2 Screening for AprY mutations related to improved thermostability
In summary, according to the preliminary screening results, S33T, T174V and N218L variants with improved thermal stability and no significant decrease in fibrinolytic activity were selected for subsequent analysis.
3.3 Purification and characterization of single point variants
In brief, (1) S33T, T174V and N218L variants exhibited unchanged optimum reaction temperature and increased thermostability; (2) The optimal pH and pH stability of all variants shifted to the alkaline region; (3) The specific enzyme activities of S33T and T174V were increased.
3.4 Biochemical characterization of rAprY variants with multiple‐point mutations
In this study, none of the designed multi-point variants achieved any further improvement in thermostability. S33T-N218L and S33T-T174V-N218L variants were inactivated; The T174V-N218L mutant lost the advantage of thermostability; Compared with the corresponding single point mutation, the t1/2 of S33T-T174V varaint was slightly increased, but it showed a balance between specific activity and short-term thermostability.
- Understanding the mechanism for enhanced thermostability and catalytic activity of nattokinase variants.
(After the overall structure analysis)In summary, the overall structure of S33T and T174V variants tended to be stable compared with rAprY, but the secondary structure composition and the number of bonds formed did not change significantly. The overall structure of S33T-T174V was more active whereas the intramolecular interactions were enhanced compared with rAprY.
(After local structure analysis)In summary, the enhanced thermostability of S33T, T174V and S33T-T174V variants is more related to their local structural changes. The flexibility changes in the active center region may be related to their enhanced catalytic activities; The improvement of rigidity of the key Loop determines the improvement of the stability of each variant; The decreased Ca2 binding ability of T174V and S33T-T174V mutants may affect the improvement of their thermostability.
In order to summarize all the experimental results, we added a Conclusion section to describe the current status of nattokinase production and application, existing problems to be solved, the results of this study, the significance of this research, and future prospects. The details are as follows:
In recent years, thrombotic diseases have become a serious threat to people's life and health. The number of deaths due to thrombotic diseases is nearly 1/4 of the total number of deaths in the world, and the data show an upward trend[2]. At present, the clinical thrombolytic drugs have side effects such as bleeding and pain during administration, and most of them are expensive. Therefore, it is urgent to seek new thrombolytic drugs with high efficiency, no side effects and low price[2]. In recent years, there have been many studies on the thrombolytic effect of nattokinase, and nattokinase health foods have also been endless. Drying nattokinase products can effectively maintain the expiry date of the product and maintain its activity, while relatively economical thermal air drying method will generate transient heat, which is detrimental to the activity of nattokinase. Current research strategies for improving the stability of nattokinase include screening of nattokinase strains, improving fermentation process, physical embedding and nanoparticle conjugation[57-59]. In addition, molecular modification using modern genetic engineering methods has gradually become an effective and economical means to improve the thermostability of nattokinase. Directed evolution and semi-rational design are commonly used methods for enzyme molecular modification, but they have the problems of high screening intensity and low positive rate. A variety of computer-aided thermal stability prediction algorithms developed at present can effectively solve the above problems, and the combination of prediction algorithms can further increase the prediction accuracy[18, 30].
In this study, two AprY variants, S33T and T174V, with enhanced thermostability and fibrinolytic activity, were successfully obtained by using several computer-aided prediction algorithms. Interestingly, the combined variant S33T-T174V did not exhibit additive thermostability effects. Using bioinformatics analysis, we found the key regions where the fibrinolytic activity and thermostability of the above mutants were increased and the reasons for the decreased thermostability of the S33T-T174V variant were also found. This study provides guidance for subsequent researchers to further enhance the catalytic properties of nattokinase, and provides excellent enzyme molecules for the food and drug development of nattokinase products, which is of great significance for the prevention and treatment of vascular embolic diseases.
In addition, in order to echo the research results, we have revised the abstract section and added some experimental results to cover the core experimental data as much as possible.
We hope that the above instructions and modifications can meet the publication requirements of the paper. Thank you again for your valuable advice.
The literature added in the revision of the paper is attached.
1, Liu, S., et al., Synthesis of sustained release/controlled release nanoparticles carrying nattokinase and their application in thrombolysis. Pharmazie, 2021. 76(4): p. 145-149.
2, Bi, J., et al., Computation-aided engineering of starch-debranching pullulanase from Bacillus thermoleovorans for enhanced thermostability. Appl Microbiol Biotechnol, 2020. 104(17): p. 7551-7562.
3, Li, G., et al., Identification of a hot-spot to enhance Candida rugosa lipase thermostability by rational design methods. RSC Adv, 2018. 8(4): p. 1948-1957.

Reviewer 4 Report
The manuscript end points, limits and scope should be better assessed and discussed also with reference to real cases possible applications in the food area. A Conclusion section should be added to clear both the limits and possible applications of the performed study in food area of interest.
The manuscript English language seems fine. Moderate language revision should fix minor issues.
Author Response
Dear professor
We quite appreciate your favorite consideration and the reviewers’ insightful comments concerning our manuscript entitled “Combined computer-aided predictors to improve the thermostability of nattokinase” (ID: foods-2537819). Those comments are very valuable and helpful for improving the quality and readability of our paper, as well as the important guiding significance to our future researches. We have studied the comments carefully and have revised the paper exactly according to the reviewers’ comments. We hope this revision can meet with approval. The revised portions are marked in red in the paper and the main revisions corresponding to the reviewers’ comments are as follows:
- The manuscript end points, limits and scope should be better assessed and discussed also with reference to real cases possible applications in the food area. A Conclusion section should be added to clear both the limits and possible applications of the performed study in food area of interest.
Thank you very much for your advice. We added a conclusion section to give an overview of the production and application of nattokinase, and summarized the conclusions of this study. Finally, we expounded the significance of the study and prospected the future research. All modified sections are marked in red and attached with added references.
In recent years, thrombotic diseases have become a serious threat to people's life and health. The number of deaths due to thrombotic diseases is nearly 1/4 of the total number of deaths in the world, and the data show an upward trend(literature 1). At present, the clinical thrombolytic drugs have side effects such as bleeding and pain during administration, and most of them are expensive. Therefore, it is urgent to seek new thrombolytic drugs with high efficiency, no side effects and low price(literature 1). In recent years, there have been many studies on the thrombolytic effect of nattokinase, and nattokinase health foods have also been endless. Drying nattokinase products can effectively maintain the expiry date of the product and maintain its activity, while relatively economical thermal air drying method will generate transient heat, which is detrimental to the activity of nattokinase. Current research strategies for improving the stability of nattokinase include screening of nattokinase strains, improving fermentation process, physical embedding and nanoparticle conjugation(literature 2-4). In addition, molecular modification using modern genetic engineering methods has gradually become an effective and economical means to improve the thermostability of nattokinase. Directed evolution and semi-rational design are commonly used methods for enzyme molecular modification, but they have the problems of high screening intensity and low positive rate. A variety of computer-aided thermal stability prediction algorithms developed at present can effectively solve the above problems, and the combination of prediction algorithms can further increase the prediction accuracy(literature 5-6).
In this study, two AprY variants, S33T and T174V, with enhanced thermostability and fibrinolytic activity, were successfully obtained by using several computer-aided prediction algorithms. Interestingly, the combined variant S33T-T174V did not exhibit additive thermostability effects. Using bioinformatics analysis, we found the key regions where the fibrinolytic activity and thermostability of the above mutants were increased and the reasons for the decreased thermostability of the S33T-T174V variant were also found. This study provides guidance for subsequent researchers to further enhance the catalytic properties of nattokinase, and provides excellent enzyme molecules for the food and drug development of nattokinase products, which is of great significance for the prevention and treatment of vascular embolic diseases.
- Yuan, L., et al., Biotechnology, Bioengineering and Applications of Bacillus Nattokinase. Biomolecules, 2022. 12(7).
- Mahajan, P.M.; Nayak, S.; Lele, S.S. Fibrinolytic enzyme from newly isolated marine bacterium Bacillus subtilis ICTF-1: Media optimization, purification and characterization.
- Zhang, X., et al., Chitosan/casein based microparticles with a bilayer shell-core structure for oral delivery of nattokinase. Food Funct, 2020. 11(12): p. 10799-10816.
- Liu, S., et al., Synthesis of sustained release/controlled release nanoparticles carrying nattokinase and their application in thrombolysis. Pharmazie, 2021. 76(4): p. 145-149.
- Bi, J., et al., Computation-aided engineering of starch-debranching pullulanase from Bacillus thermoleovorans for enhanced thermostability. Appl Microbiol Biotechnol, 2020. 104(17): p. 7551-7562.
- Li, G., et al., Identification of a hot-spot to enhance Candida rugosa lipase thermostability by rational design methods. RSC Adv, 2018. 8(4): p. 1948-1957.
As a further supplement to your suggestion, we have added the production and application of nattokinase products, the method of nattokinase activity determination, and the theoretical content of nattokinase molecular modification in the results and discussion section, respectively.
At the same time, we have made an adjustment to highlight the application of nattokinase in the food field in the abstract and introduction.
- The manuscript English language seems fine. Moderate language revision should fix minor issues.
Thank you for your suggestions. We have noticed some grammatical errors in the article and have made corrections.

Round 2
Reviewer 1 Report
Authors have improved the manuscript based on the comments and it can be accepted now for publication.